# Using machine learning as a surrogate model for agent-based simulations

Claudio Angione[1,3,4,5]☯*, Eric Silverman[2]☯, Elisabeth Yaneske[1]☯

**1** School of Computing, Engineering and Digital Technologies, Teesside University, Middlesbrough, United Kingdom, **2** Institute for Health and Wellbeing, University of Glasgow, Glasgow, United Kingdom, **3** Healthcare Innovation Centre, Teesside University, Middlesbrough, United Kingdom, **4** National Horizons Centre, Teesside University, Darlington, United Kingdom, **5** Centre for Digital Innovation, Teesside University, Middlesbrough, United Kingdom

☯ These authors contributed equally to this work.
* c.angione@tees.ac.uk

**Data Availability Statement:** The code and the results presented in this manuscript are shared in full in our GitHub repository https://github.com/thorsilver/Emulating-ABMs-with-ML. Our simulation of the 'Linked Lives' ABM model of

## Abstract

In this proof-of-concept work, we evaluate the performance of multiple machine-learning methods as surrogate models for use in the analysis of agent-based models (ABMs). Analysing agent-based modelling outputs can be challenging, as the relationships between input parameters can be non-linear or even chaotic even in relatively simple models, and each model run can require significant CPU time. Surrogate modelling, in which a statistical model of the ABM is constructed to facilitate detailed model analyses, has been proposed as an alternative to computationally costly Monte Carlo methods. Here we compare multiple machine-learning methods for ABM surrogate modelling in order to determine the approaches best suited as a surrogate for modelling the complex behaviour of ABMs. Our results suggest that, in most scenarios, artificial neural networks (ANNs) and gradient-boosted trees outperform Gaussian process surrogates, currently the most commonly used method for the surrogate modelling of complex computational models. ANNs produced the most accurate model replications in scenarios with high numbers of model runs, although training times were longer than the other methods. We propose that agent-based modelling would benefit from using machine-learning methods for surrogate modelling, as this can facilitate more robust sensitivity analyses for the models while also reducing CPU time consumption when calibrating and analysing the simulation.

## Introduction

In this paper, we investigate the use of machine-learning-based surrogate modelling for the analysis of agent-based models (ABMs). In this approach, machine-learning methods are used to generate statistical models that replicate the behaviour of the original ABM to a high degree of accuracy; these surrogates are substantially faster to run than the original model, enabling complex sensitivity analyses to be performed much more efficiently. This proof-of-concept work demonstrates that these methods are applicable and useful even in time- and resource-limited modelling contexts, and that these surrogates are capable of closely replicating the

social care provision in the UK is available at https://github.com/thorsilver/Social-Care-ABM-for-UQ/releases/tag/v0.91.

**Funding:** ES is part of the Complexity in Health Improvement Programme supported by the Medical Research Council (MC_UU_00022/1) and the Chief Scientist Office (SPHSU16). CA would like to acknowledge the support from UKRI EPSRC through a Turing Network Development Award, UKRI Research England's THYME project, and the Children's Liver Disease Foundation. This work was supported by UK Prevention Research Partnership MR/S037594/1, which is funded by the British Heart Foundation, Cancer Research UK, Chief Scientist Office of the Scottish Government Health and Social Care Directorates, Engineering and Physical Sciences Research Council, Economic and Social Research Council, Health and Social Care Research and Development Division (Welsh Government), Medical Research Council, National Institute for Health Research, Natural Environment Research Council, Public Health Agency (Northern Ireland), The Health Foundation and Wellcome. The funders had no role in study design, data collection and analysis, decision to publish, or preparation of the manuscript.

**Competing interests:** The authors have declared that no competing interests exist.

behaviour of the original model even when minimal hyperparameter optimisation is performed. We propose that incorporating such methods into standard agent-based modelling practice may allow a significant improvement in the standard of results reporting in certain disciplines, particularly in policy-relevant contexts where analyses of models frequently must be performed quickly and with limited computational resources.

Agent-based modelling is a computational modelling approach most often applied to the study of complex adaptive systems [1]. ABMs typically represent individuals directly, and situate these *agents* in a virtual environment of some kind. These agents then engage in varied behaviours encoded in a set of *decision rules* that drive their actions in response to behavioural, environmental and social change. The resultant complex interactions between agents and their environments can lead to *emergent* behaviours, in which the patterns of behaviour seen at the population level have new properties that are not straightforwardly attributable to individual-level actions [2].

Agent-based modelling is growing in popularity in social and health sciences, and recent papers have proposed that agent-based modelling has the potential to provide insight into complex policy challenges that have been resistant to traditional statistical modelling approaches [3]. However, the use of agent-based modelling presents technical barriers in implementation, and the analysis of ABM outputs is a challenging undertaking, and often very demanding of computational resources [4].

The ability of ABMs to model complex interactions and to demonstrate emergence has meant that agent-based modelling is particularly relevant to those disciplines of the social sciences where individual agency is considered important to population-level outcomes. This is not a new phenomenon; one of the very first ABMs was a social model—a simple model of residential housing segregation designed by Thomas Schelling [5]. Since the 1980s and Axelrod's *The Evolution of Cooperation* [6], this synergy with the social sciences has led to the development of the field of *social simulation*, in which this variety of computational social science is used to examine the development and evolution of human society in a wide variety of circumstances [7]. In recent years, more applied areas of social science, such as public health, have proposed agent-based modelling as a means to investigate societal responses to new social or economic policies [3].

As agent-based modelling becomes more commonplace in policy debates, methodological discussions amongst agent-based modelling practitioners have focused on the development of more efficient means for understanding the outputs of these complex simulations. Even a simple ABM may have several interacting processes affecting its agents, meaning that the relationships between model parameters can be highly *non-linear*—small changes to one parameter can lead to unexpectedly large effects on simulation outcomes, or vice versa. This, in turn, means that understanding an ABM is a complex undertaking, often involving detailed sensitivity analyses performed using Monte Carlo methods, where large numbers of repeated simulation runs are performed at a wide range of parameter values. When ABMs are highly complex, performing these kinds of analyses becomes both time- and cost-prohibitive, potentially leading some modellers to truncate these analyses or eliminate them entirely, leading to a model that is less robust to changes in parameter values [8].

When such analyses are not performed, model verification and validation can become more challenging. In this context, sensitivity analyses can be used to provide insights into the contributions made by model parameters, which can be valuable for verifying whether the model processes driven by those parameters are functioning correctly. ABMs simulate multi-level complex systems, which may be multi-realisable, meaning that the macro-level phenomena of interest may be generated in multiple ways at the individual level [9]. That being the case, validation for ABMs can be particularly challenging, as the ABM may be only one of

many possible explanations for the macro-level phenomena. Sensitivity analysis can therefore provide a more complete description of the model's behaviour, which can help the modeller to justify why their implementation constitutes a useful explanation of the underlying system. Without such analyses, validation relies on face validity and on matching input-output transformations to real data, which cannot provide evidence on whether the underlying model processes and assumptions are valid.

In addition, in real health and social policy scenarios, decisions are often taken under significant pressure, and in very short timeframes. Such decisions are often high-risk, affecting millions of individuals, and may generate negative outcomes [10]. Competing political interests and economic pressures have a strong influence on the outcomes of the policy-making process [11]. For modelling tasks within such policy development process, this may create an environment where detailed interrogation of the models may not be practical. Therefore, developing model analyses that are practical in these environments would benefit from the use of alternative methods, rather than the conventional Monte Carlo approach.

Recent developments in *uncertainty quantification (UQ)* have provided alternative means for calibrating and analysing complex simulation models. Using methods like *surrogate modelling* allows creating a statistical model of the simulation model, meaning the repeated simulation 'runs' can be completed in mere seconds using a statistical surrogate of the original complex ABM [12, 13]. The most common approach is *Gaussian process emulation (GP)*, which has been used with some success in agent-based modelling applications in a variety of fields including public health [14, 15]. We note that the terms 'surrogate model' and 'emulator' are sometimes used interchangeably, but the originators of the GP approach use the term 'emulator' specifically for methods that provide full probabilistic predictions of simulation behaviour, not only approximations of the outputs.

Importantly, surrogates cannot serve as complete *replacements* for the original ABM; the goal of surrogate modelling is to complement the modelling process and help illuminate the behaviour of the original model. Surrogate models can reduce the significant computational demands of the calibration and sensitivity analysis processes, but if one wishes to test the simulation and its assumptions on new empirical information, then the original model should be used, not the surrogate model. However, even in this limited role, the use of surrogate models can save significant computational resources, given that a typical surrogate model is many orders of magnitude faster to run than a complex computational simulation [16].

At the same time, machine and deep learning approaches have shown wide applicability and versatility when simulating mechanistic processes, merging model-driven and data-driven techniques [17]. In agent-based modelling, machine learning approaches have been used for two main applications [18]: (i) modelling adaptive agents that can learn from experience through reinforcement learning approaches; and (ii) analysing and post-processing the (often large-scale) outcomes of running a given ABM. With the advent of accessible machine-learning methods including *artificial neural networks (ANNs)*, various authors have also proposed machine learning as a potentially productive means of creating surrogate models for ABMs [19, 20].

These frameworks seem particularly promising in the case of ANNs; while it is well-known that shallow-depth ANNs are universal approximators capable of approximating any continuous function, recent work has shown that modern deep neural networks are capable of similar or greater expressiveness even when the network width is limited [21]. These properties suggest that ANNs can more easily handle the highly non-linear nature of ABMs compared to other approaches. Other machine-learning approaches are significantly faster than training an ANN, and may also prove fruitful for surrogate modelling purposes.

Despite its high potential, machine learning can be found only in a very limited number of applications in the agent-based modelling domain. As well as having been used as part of an agent-based modelling calibration process [22, 23], one recent project has proposed to determine single-agent behaviour using machine learning. In this project, a machine-learning model was trained to mimic behavioural patterns, where parameters are given as input and the action is the output, all performed at a single-agent resolution [24]. At the time of writing, machine-learning methods have not been used widely to develop surrogate models for the purpose of sensitivity analysis. The majority of extant examples of machine-learning methods applied to ABM outputs are GP implementations. To our knowledge, various versions of ANNs and other methods have been discussed in principle, but not yet designed and implemented on agent-based simulations applied to human social systems.

In this paper, we test a range of machine-learning approaches for the task of replicating the outputs of an ABM that has been used to evaluate social care policies in the UK [14], in order to investigate the potential of these approaches for generating surrogate models of complex policy-relevant models. The model was chosen as an exemplar of the type of ABM that may be applied in policy-relevant modelling studies. We note that models vary widely in complexity, empirical relevance, and underlying behavioural assumptions; therefore, this model serves as a case study, and we do not claim these results will apply to all ABMs. We also apply the chosen machine-learning methods with minimal hyperparameter optimisation, and perform all our calculations on commodity hardware, in order to test the applicability of these methods in conditions reflective of the constraints and optimality tradeoffs that policy-relevant agent-based modelling work is required to meet.

We propose that machine and deep learning methods, when applied to the generation of surrogate models, can improve the theoretical understanding of the ABM, help calibrate the model more efficiently, and provide more insightful interpretations of the simulation outputs and behaviours. Therefore, for parameter spaces that cannot be searched effectively with heuristics, machine learning models can be learned from ABM outputs, and machine/deep learning techniques can then be used as a surrogate model of the ABMs with high accuracy. We also contend that such methods can be used even in high-pressure, high-risk environments like health and social policy-making, given that they may be applied quickly and demonstrate highly accurate surrogate modelling performance even with limited hyperparameter optimisation. As this proof-of-concept work demonstrates, machine-learning-based surrogate models can replicate the behaviour of ABMs even where sophisticated methods like GP emulation have failed, suggesting that surrogate modelling may be used practically and straightforwardly even when the model's behaviour is highly complex.

## Interpretability of machine-learning models

The widespread use of machine-learning models today has led to concerns being raised regarding their interpretability, given that understanding the predictions produced by machine-learning models is far from straightforward [25]. Deep-learning models in particular are enormously complex, often containing hundreds of layers of neurons adding up to tens of millions of parameters. Recently, significant progress has been made in developing tools for interpreting these models, including recent striking interactive attempts to make a large deep-learning model interpretable [26, 27]. In biomedicine, for instance, efforts towards interpretability are paramount also when the input data is collected from different sources and is therefore inherently multimodal [28]. Such tools are progressing rapidly, but they still require a significant time investment, and are not yet in widespread use. However, we note that while large machine-learning models may appear particularly problematic, simpler methods like

logistic regression can be equally difficult to interpret when dealing with large data sets, and regularisation methods should be used to mitigate this issue [29, 30].

While machine-learning models can suffer from difficult interpretability, in the case of building surrogate models the primary aim is to significantly reduce the time required to produce model outputs for sensitivity analyses. Therefore, convenient implementation and computational efficiency are also of high importance. Notably, the surrogate model can produce analyses that help to understand the behaviour of the original model, regardless of the interpretability of the surrogate itself.

Machine learning techniques, if used to mimic ABMs and coupled with tools for interpretations of the predictions, can shed light on the input-output relationships. In this context, the recent strong focus on explainability techniques for machine learning models, e.g. SHAP [26], could also quantify the influence that each factor has on the model outcome. For a more in-depth discussion on the advantages and disadvantages of these approaches in the context of ABMs, we refer the reader to the recent systematic literature review and discussion by Dahlke et al. [18].

In this paper, we implement and provide a thorough comparison of the performance of a multitude of machine-learning methods in an ABM surrogate modelling scenario. As an exemplar model, we have used an agent-based model of social care provision in the UK [15], generated sets of observations across ranges of input parameter values designed for optimum parameter space coverage, and attempted to replicate the ABM outputs using machine-learning algorithms. In the following section, we outline the methods we studied, and in the Results section we provide a detailed analysis of their performance in this task.

## Motivations

With this paper, we investigate a question raised elsewhere [19, 23]: whether neural networks and other machine-learning methods may be used successfully and efficiently as a method for surrogate modelling of ABMs. The work done in this area thus far has proposed this possibility, but has not taken the step of directly comparing machine learning to other methods for surrogate modelling of ABMs. Previous work by O'Hagan has suggested that neural networks were less well-suited to emulation tasks, by "not allowing for enough uncertainty about how the true simulator will behave" [13]; however, since that paper was written in 2006, neural network approaches have advanced significantly. In an effort to spur further work on this topic, we have developed this comparison of nine different possible methods, with a particular focus on examining the potential for using neural networks for surrogate modelling.

We have chosen a selection of the most widely-used machine learning methods for our comparison. The primary advantage of using surrogate models is to drastically reduce the time required to perform detailed analyses of ABM outputs; with that in mind, we chose methods that can generate predictions rapidly once the surrogate is trained. These methods can differ significantly in the time required for training the model, so we compare training times in this paper as well as predictive accuracy. We attempted to provide a survey of numerous ML methods, discussing their strengths and weaknesses specifically for potential use as surrogates. While our case study clearly shows differences in the performance across the machine learning methods and one can in principle pre-select the best-performing methods, we provided the complete set of results as the best method might in principle differ depending on the specific ABM under investigation, and its complexity.

Substantial work remains to be done on developing these methods into a more accessible form aimed specifically at model calibration and surrogate modelling, and to further refine their application in this context. We aim to inform these efforts, and to determine whether

useful surrogate models can be built using machine learning methods, and which methods are most effective.

We also address the problems ABMs pose for current surrogate modelling methods, such as Gaussian process emulators, and investigate whether machine-learning methods can surpass these obstacles. We therefore included Gaussian process emulators in our comparative framework. Both GP emulation and Kriging are popular forms of Gaussian process regression. Here we opted to focus on GP emulation as the main comparator, because GP emulation is found more often in the ABM literature. As we demonstrate below, Gaussian process emulators are both efficient and powerful, but can struggle nonetheless to fit the output of even moderately complex ABMs. Our hope is that ultimately new methods for surrogate modelling can supplement GPs in these cases.

To ensure our results are accessible to a wide range of modellers, we have performed our analyses using Mathematica, Matlab and R. The code and results are shared in full in our GitHub repository https://github.com/thorsilver/Emulating-ABMs-with-ML. We chose this approach in the hope that even modellers familiar with graphical agent-based modelling tools like NetLogo or Repast will be able to try the methods and reproduce our results even without extensive programming experience. We performed all our analyses on mid-range desktop PCs, and did not use GPUs for any calculations; this was to ensure a fair comparison across all methods, and that our models could be built even without high-performance computing hardware. Similarly, we performed hyperparameter optimisation to provide a realistic picture of how each method would perform in circumstances where optimality tradeoffs and time constraints are likely required. We note that methods like artificial neural networks may produce even stronger results when extensively optimised.

In order to properly interrogate the results, there are currently no universally applied standards regarding how ABMs should be calibrated and analysed. Therefore, in time- and resource-limited contexts, more complex analyses of sensitivity and uncertainty are sometimes omitted to save time and computational expense. We believe that machine learning surrogates that can be built within such constraints provide robust and accessible ways to analyse ABMs even in challenging modelling contexts. We propose that such tools could significantly improve the standard of results reporting by making the production of detailed sensitivity analyses more straightforward and less computationally costly, and this, in turn, may widen the use of agent-based modelling studies in other areas. Our aim is to produce proof-of-concept work that illustrates this possibility, and inspires future development of accessible frameworks for automating machine-learning-based surrogate modelling for ABMs.

## Materials and methods

Table 1 provides a summary of the machine learning methods studied in this paper, and includes a description of each method, with its advantages and disadvantages. This summary is provided as a quick guide, rather than as a definitive comparison between them. For further details on each method, we refer the reader to the key references cited in each description.

## The 'Linked Lives' agent-based model

To present a useful comparison of methods for generating surrogate ABMs, we chose to use an exemplar ABM which is neither a simplistic 'toy model', nor a highly-detailed simulation with hundreds of parameters. Simplistic ABMs may generate behaviour that is easier to replicate, i.e. they would not present a sufficient test of the capabilities of the emulators, and in practice some simplistic models could be analysed relatively effectively using traditional methods, given the short runtimes. Conversely, high-complexity ABMs may provide a very robust test

**Table 1. Summary of methods implemented in our study.**

| Method | Description | Advantages | Disadvantages |
|---|---|---|---|
| Linear Regression | Predicts values using a linear combination of input features, in this case parameter values [31]. | Fast, extremely well-studied, easy to implement with any number of tools. | Not well-suited for data with non-linear relationships. |
| Decision Trees | Generates binary trees that predict the value of a variable based on several inputs, represented as interior nodes in the tree [32]. | Fast and easy to train, simple to implement, very easy to understand and interpret. | High variance (slight changes to input data can produce very different trees), prone to overfitting. |
| Random Forests | Ensemble approach to decision trees in which multiple models are trained using random split points. When making a prediction, the predictions of each tree in the ensemble are averaged together to produce the final result [33]. | Easy to parallelise, low computational load, low variance. | Prone to overfitting, low interpretability. |
| Gradient Boosted Trees | Ensemble method that combines weak learners into a strong learner [34]. Weak learners are trained sequentially so that each successive learner improves on its predecessor's predictions. XGBoost [35] and LightGBM [36] have further optimised this approach and are frequently used in a variety of machine-learning domains. | Fast, low variance, very successful across a wide range of problems. | Prone to overfitting, requires extensive parameter optimisation. |
| K-Nearest Neighbours | This method makes a prediction by finding the K most similar points in the entire training data set to the new data point we would like to label, then summarising the output values at those points to arrive at the prediction for the new point [37]. | Simple to implement, fast, only requires computational resources when making a prediction. | Must include the entire training set, becomes ineffective as the number of input variables becomes high (the 'curse of dimensionality'). |
| Gaussian Process Emulation | The most common method for surrogate modelling of computer models, GPs model the simulation as a Gaussian process [13]. Useful measures like the *main effects*, or output variance due to each input parameter, are straightforwardly derivable from the emulator. | Very computationally efficient, highly useful for sensitivity analyses, specialised free software (GEM-SA) speeds the process up considerably. | Assumes that the surrogate model is smooth (may not be the case with complex ABMs), GEM-SA software no longer maintained, only copes with single model outputs. |
| Support Vector Machine (SVM) | Finds a hyperplane in a high-dimensional space of data points that can separate those points with the widest possible margin. It can be used for classification or regression [38]. | Scales well, few hyperparameters to optimise, flexible and powerful in higher dimensions thanks to the 'kernel trick'. | It can be difficult to choose the right kernel, long training times for large datasets, low interpretability. |
| Neural Network | A network of nodes loosely based on biological neurons, normally consisting of an input and output layer with one or more hidden layers of neurons in between. Learning algorithms adjust the weighted connections between neurons to enable regression or classification of input datasets. Deep neural networks use many hidden layers and can model complex non-linear relationships [39]. | An enormous variety of possible layer types and network architectures, can learn supervised or unsupervised, highly suitable for modelling non-linear relationships, very well-supported by powerful open-source software. | Computationally expensive hyperparameter optimisation, prone to overfitting, large networks require GPU access for training, low interpretability. |

of the capabilities of ML-based emulation. However, a highly complex model would have taken impractical amounts of CPU time to generate output data and would not be representative of the typical empirical ABM, where overly complex models are often avoided to enable the model to provide explanatory insight into the system of interest. Therefore, for reasons of applicability, practicality, and replicability we chose to use a model between these two extremes, which we call 'moderately complex', with runtimes on the order of several minutes and ten configurable parameters.

The chosen model is the 'Linked Lives' model of social care provision in the UK, which models the interaction between supply and demand of informal social care [15, 40]. The model is written in Python, and simulates the movements and life course decisions of agents

in a virtual space built as a rough approximation of UK geography. Our simulation code is freely available via GitHub, and the specific release used for this paper is available at https://github.com/thorsilver/Social-Care-ABM-for-UQ/releases/tag/v0.91.

The Linked Lives model provides a platform for the investigation of social and economic policies directed at formal and informal UK social care. Social care in the UK is a crucial policy question, as insufficient care being provided can lead to vulnerable people needing hospital treatment or their health status declining further. The care supply in the UK is insufficient for the demand, and demand is projected to rise in the coming decades due to the current demographic trends [41, 42]. A significant proportion of care is provided by informal carers, typically family members who provide their time free of charge to assist one of their relations with daily activities [8, 43]. The Linked Lives model aims to provide a representation of the tradeoffs faced by informal carers as they make decisions about care provision, and thus to enable policy-makers to better address the particular needs of informal carers.

The model includes a rough representation of UK geography, in which clusters of households form into cities that mirror the population density of the UK. As the simulation progresses, some agents will develop a need for social care due to long-term health conditions, with the amount of care required varying according to the agent's level of need. Family members of that agent will attempt to provide social care, but the amount of care they can provide is affected by the social and economic conditions in which they live. The model thus simulates the varied and complex factors that influence an individual's ability to provide informal care to their loved ones.

Agents in the Linked Lives model are capable of migrating domestically, forming and dissolving partnerships, and participating in a simple simulated economy. As agents age, they may develop long-term limiting health conditions that require social care; the probability of developing such a condition varies by age and gender. Care availability varies according to a potential carer's employment status, geographical location and health status. These caregiving patterns may shift in response to policy changes, which can be implemented in the simulation by altering the simulation parameters.

The Linked Lives model is more complex than theoretical ABMs built to examine foundational aspects of social behaviour, such as Schelling's residential segregation model [5], and better reflects the level of detail seen in empirically-informed models aimed at simulating real-world social systems. The model is intended as a tool for policy development and evaluation in social care, an area of social policy where policy changes may affect millions of families. In this context, an in-depth understanding of the model behaviour is essential if policy-makers need to trust the simulation as a decision support tool. With that in mind, we have chosen the Linked Lives model for this study because it serves as an example of the models being constructed with increasing frequency in various areas of population health. Unlike 'toy models' investigating social theory, we decided to adopt a real-world policy-relevant tool informed by empirical data, to be used for interrogating real-world complex systems.

## Properties of the Linked Lives model

The Linked Lives model would be classified in the emulation literature as a *deterministic simulator*, meaning that a given set of parameter values will produce the same results, presuming that the same random number seed is used. Many simulations are of this type, as they will be implemented in standard programming languages which use pseudo-random number generators.

In previous explorations, the Linked Lives model has demonstrated *ergodicity*, i.e. the simulation tends to converge to similar outputs at identical parameter values, even if the random

seed is different. However, we note that this property, while useful in that it suggests less code uncertainty is present in the simulation, is not essential to the use of surrogate modelling techniques. If the random seed has a significant influence on model outputs, surrogates like Gaussian process emulators can still be constructed. In this context, the distinction between stochastic and deterministic models can be sometimes artificial, as a stochastic simulator can be interpreted as deterministic when the random number seed becomes an input [13].

Therefore, when generating runs for a simulation that has a high degree of stochasticity, modellers should record the used random seed with each run to enable the inclusion of those seeds as an additional input when training the surrogate. In this way the simulation can be treated as a deterministic model.

## Generating simulation data

The Linked Lives model contains 22 user-alterable parameters, 10 of which are potentially relevant to modelling social care policies. Table 2 summarises the ten parameters and their function within the simulation. The input values for these parameters were varied across ranges determined by experimentation to lie at the upper and lower bounds for interpretable simulation behaviour; beyond those bounds, the simulation results were generally highly unrealistic, leading to either collapsing or exploding agent populations.

The simulation output of interest is the per capita cost of social care per annum. Each simulation run starts with a random initial population in the year 1860, then runs in yearly time steps until 2050. The final per capita cost of care in 2050 is then recorded along with the input parameter values for that run. Full simulation runs were generated in four different batches consisting of 200, 400, 800 and 1600 runs, in order to allow us to compare the performance of each machine learning method with both smaller and larger sets of observations. Each run constituted a sample for the machine learning methods.

When building a surrogate model, generating an appropriate experimental design is essential; the runs of the original simulation must be therefore conducted such that they cover a sufficient portion of the mode's parameter space. In the Gaussian process emulation literature, maximin Latin Hypercube Design [44] and LP-tau [45] are most typically used to generate experimental designs, and both of these methods are available in the widely-used GEM-SA software package for Gaussian process emulation [46]. The LP-tau method enables the rapid generation of experimental designs even in complex parameter spaces, and provides a good uniformity of distribution across the parameter space [47]. In keeping with our aim to demonstrate the utility of surrogate modelling approaches even in time- and resource-limited

**Table 2. The ten parameters used in the Linked Lives ABM surrogate model generation process, with descriptions, default values and lower and upper bounds used when generating simulation output data.**

| Parameter | Description | Default | Range |
|---|---|---|---|
| ageingParentsMoveInWithKids | Probability agents move back in with adult children | 0.1 | 0.1–0.8 |
| baseCareProb | Base probability used for care provision functions | 0.0002 | 0.0002–0.0016 |
| retiredHours | Hours of care provided by retired agents | 60.0 | 40–80 |
| ageOfRetirement | Age of retirement for working agents | 65 | 55–75 |
| personCareProb | General individual probability of requiring care | 0.0008 | 0.0002–0.0016 |
| maleAgeCareScaling | Scaling factor for likelihood of care need for males | 18.0 | 10–25 |
| femaleAgeCareScaling | Scaling factor for likelihood of care need for females | 19.0 | 10–25 |
| childHours | Hours of care provided by children living at home | 5.0 | 1–10 |
| homeAdultHours | Hours of care provided by unemployed adults | 30.0 | 5–50 |
| workingAdultHours | Hours of care provided by employed adults | 25.0 | 5–40 |

modelling contexts, we elected to use LP-tau for the generation of our experimental design. The parameter ranges listed in Table 2 were used to generate the four experimental designs using LP-tau, and the simulation was then run at the generated parameter values for each of those four designs. These four designs spanned the parameter space using 200, 400, 800 and 1600 runs, meaning that over the course of this phase of the study we ran the simulation a total of 3,000 separate times at different parameter values. Note that the LP-tau method was used to generate the parameter values for each scenario separately, to ensure that the parameter space coverage was distributed uniformly in each case; as a consequence, each scenario used different parameter values for each individual run.

## Machine learning models to predict simulation outputs

For each of the machine learning algorithms chosen for this comparison, the simulation outputs for each of the four batches outlined above were used to train each algorithm to predict the output of the simulation. The ten simulation parameters described in Table 2 were used as input features, and the output to be predicted is the social care cost per capita per annum.

The loss function used was the mean-squared error (MSE), the most commonly-used loss function for regression tasks. This is calculated as the mean of the squared differences between actual and predicted values:

$$MSE = \frac{1}{n}\sum_{i=1}^{n}(\hat{y}_i - y_i)^2,$$

(1)

where $\hat{y}$ is the $n$-dimensional vector of actual values, and $y$ is the $n$-dimensional vector of predicted values. Training times were recorded in units of seconds for each algorithm.

All the tested machine learning methods were trained by splitting randomly the simulation run data into training, validation and test sets, as commonly carried out in machine learning approaches. In our simulation, the test set consisted of 20% of the initial set of runs, the validation set consisted of 20% of the remaining 80% after the test set was created, and all the remaining data formed the training set. This three-way split allowed for hyperparameter optimisation to be done using the validation set, without any risk of accidentally training a model on the test set and thus obtaining biased results. The training set was used exclusively for training, and the validation set for the evaluation of the trained model. The final-round MSE loss figures on the test set were then compiled into our results table (within Fig 1), providing a summary of the relative performance of all the machine learning methods for creating surrogate models of the ABM.

## Considerations on creating ABM surrogate models with machine learning algorithms

Once the simulation runs were completed, we chose a selection of the most commonly used machine learning methods to evaluate as possible means for creating surrogate models of the ABM. Given that the motivation for generating these surrogates is to enable detailed analysis of ABM outputs without having to run the ABM many thousands of times, we sought to test methods that would produce predictions very quickly once the model is trained. As part of our comparison table in Fig 1, we included the training time required for each surrogate model.

For this comparison, we investigated the performance of these approaches without requiring a pipeline of extensive hyperparameter optimisation. We note that some of these methods, and especially neural networks, could produce an even stronger result in both accuracy and speed by exploiting recent advances in deep learning surrogates [16, 48, 49].

| | 200 Runs | | 400 Runs | | 800 Runs | | 1600 Runs | |
|---|---|---|---|---|---|---|---|---|
| | Runtime | MSE Loss | Runtime | MSE Loss | Runtime | MSE Loss | Runtime | MSE Loss |
| Neural Network | 47.5s | 0.965 | 91.77s | 0.761 | 222.97s | 0.224 | 695.31s | 0.188 |
| Linear SVM | 1.42s | 6.246 | 2.4s | 7.023 | 3.4s | 7.826 | 11.17s | 3.22 |
| Non-Linear SVM | 1.7s | 5.254 | 2.87s | 2.586 | 5.61s | 0.708 | 14.25s | 0.236 |
| Random Forest | 2.14s | 16.936 | 0.397s | 12.06 | 1.58s | 2.974 | 2.16s | 2.656 |
| Linear Regression | 1.47s | 5.237 | 1.42s | 4.374 | 3.06s | 13.439 | 3.86s | 24.18 |
| Gradient Boosted Trees | 1.4s | 3.778 | 1.47s | 2.526 | 1.5s | 0.452 | 3.88s | 2.774 |
| K-Nearest Neighbours | 0.33s | 12.895 | 0.35s | 13.422 | 0.37s | 1.356 | 2.29s | 7.404 |
| Gaussian Process | 2.02s | 4.284 | 2.97s | 1.928 | 4.12s | 0.33 | 17.7s | 37.02 |
| Decision Trees | 0.33s | 19.734 | 0.355s | 14.321 | 0.532s | 3.855 | 2.09s | 2.247 |

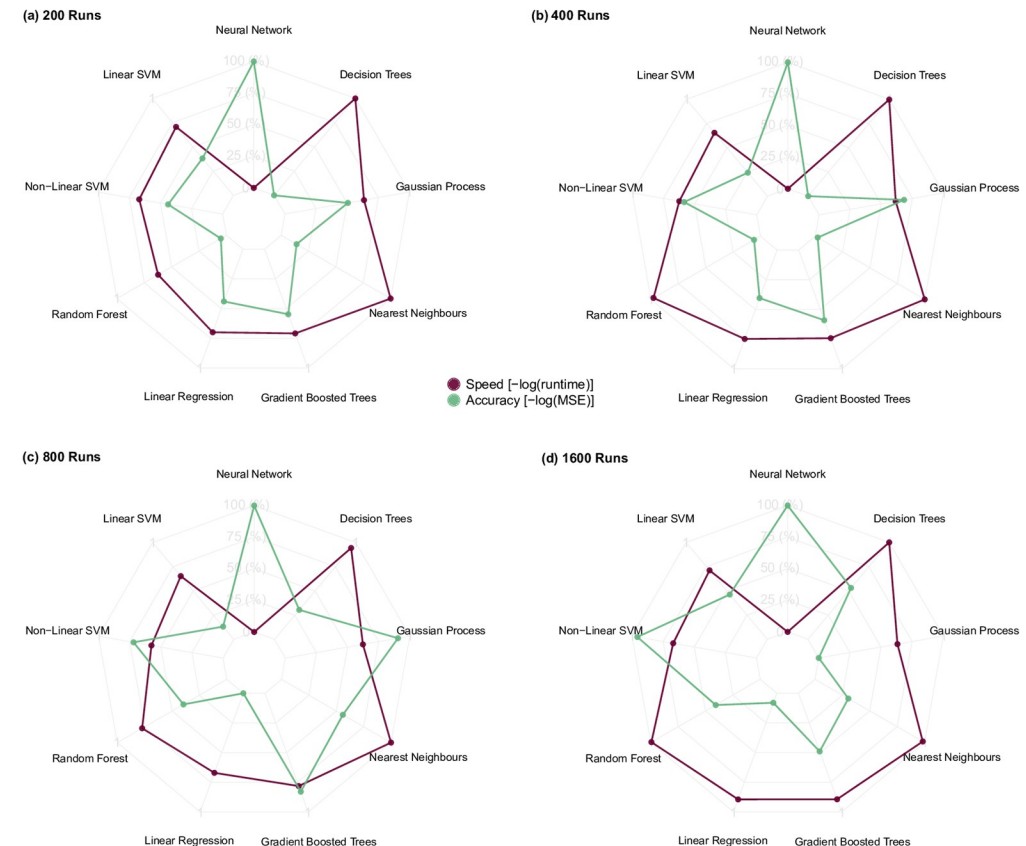

**Fig 1. Performance of the nine machine-learning methods trained on simulation outputs from 200, 400, 800 and 1600 runs.** The spider plots compare speed and accuracy across all nine methods for the 200, 400, 800 and 1600 run scenarios in plots (a), (b), (c) and (d) respectively. For each method, the total computational runtime on an 8-core i7 CPU and the mean-squared error (MSE) on the test set are shown (both in log scale, reversed, and mapped to the [0, 1] interval to represent relative speed and accuracy, respectively). Neural networks were the strongest overall performers, with gradient-boosted trees also performing well overall, and non-linear SVM performing increasingly well for higher numbers of runs. The high accuracy of the neural network models has a significant cost in terms of speed. Gradient-boosted trees and non-linear SVM consistently perform well in terms of speed, but suffer from a lower accuracy overall.

We note that in addition to the methods listed, we also tested XGBoost [35] and LightGBM [36]. However, both of these methods proved unable to generate a usable model of the ABM data. This may be due to these methods being more specialised for modelling very large input datasets. Our simulation datasets are comparatively small, and in this study we are seeking methods for generating surrogate models that can provide good predictions even with small training sets, thus allowing substantial time saving when performing model calibration and analyses. That being the case, we elected not to report the results for XGBoost and LightGBM in this paper, as neither method proved optimal for this specific purpose.

All the methods listed in Table 1, except for support vector machines (SVM) and neural networks, were implemented using Mathematica 12.0 [50], which has a comprehensive machine learning suite included. The split for training, validation and testing was fixed for all the methods. Mathematica's *Predict* function was then used to produce a model based on the supplied input training data, which was then analysed. SVM was implemented in R, while neural networks were built with the *NetTrain* function, as detailed in the sections below.

**Support vector machine implementation.** Linear and non-linear SVM were implemented in RStudio v1.3.1093 using the *caret* package. The non-linear model was trained using the radial basis function kernel, while the linear model was trained using the linear kernel function. In the case of SVM, the data was randomly allocated into the training, validation and test sets for each sample.

**Artificial neural network implementation.** The neural network models were implemented using Mathematica's *NetTrain* function. The networks were deep neural networks with a 10-node input layer, batch normalisation layer, *tanh* layer, a variable number of fully connected hidden layers, batch normalisation layer, *tanh* layer, and a single-node output layer (outputting a scalar). For each set of runs, a brief grid search was conducted using steadily increasing numbers of hidden layers until we found an architecture that provided the best possible fit without overfitting. This produced neural networks that increased in depth as the size of the training set increased. The learning rate was set at 0.0003 for all networks, with L2 regularisation set at 0.03 to help avoid overfitting. All networks were trained for 15,000 epochs, with batch sizes left to default values.

Fig 2a shows a sample architecture for the most successful neural network for the 800-run training set. This network contains a total of 13 layers, nine of those being fully connected hidden layers with 50 nodes each. Fig 2b shows a simplified view of the network structure. Fig 2c shows the loss decreasing with the training epochs for a sample neural network trained on the 800-run simulation dataset. The actual and predicted test values are shown in Fig 2d. All the code and results are available in our Github repository: https://github.com/thorsilver/Emulating-ABMs-with-ML.

## Results

### Initial investigation using GPs

Following the generation of our simulation results, we performed an initial uncertainty analysis using a Gaussian process emulator (GP). This analysis was performed using the GEM-SA software package [13, 46], using data gathered from 400 simulation runs. The emulator output in Fig 3 shows that the GP emulator was unable to fit a GP model to the simulation data, suffering from an extremely large output variance.

GPs are limited in their capacity to be a surrogate model of certain kinds of complex simulation models, in that they assume that simulation output varies smoothly in response to changing inputs [12, 13]. In the case of complex ABMs, this assumption often fails as model outputs can change in unexpected ways in response to small variations in input parameter values. Our exemplar ABM falls into this category, meaning that even though other critical GP assumptions still hold (the model is deterministic, and has a relatively low number of inputs), the emulator is unable to fit a GP model to the simulation data. As the next analysis will show, in this case, determining the impact of the input variables on the final output variance is challenging. This may also have contributed to the low performance of this initial GP emulator attempt.

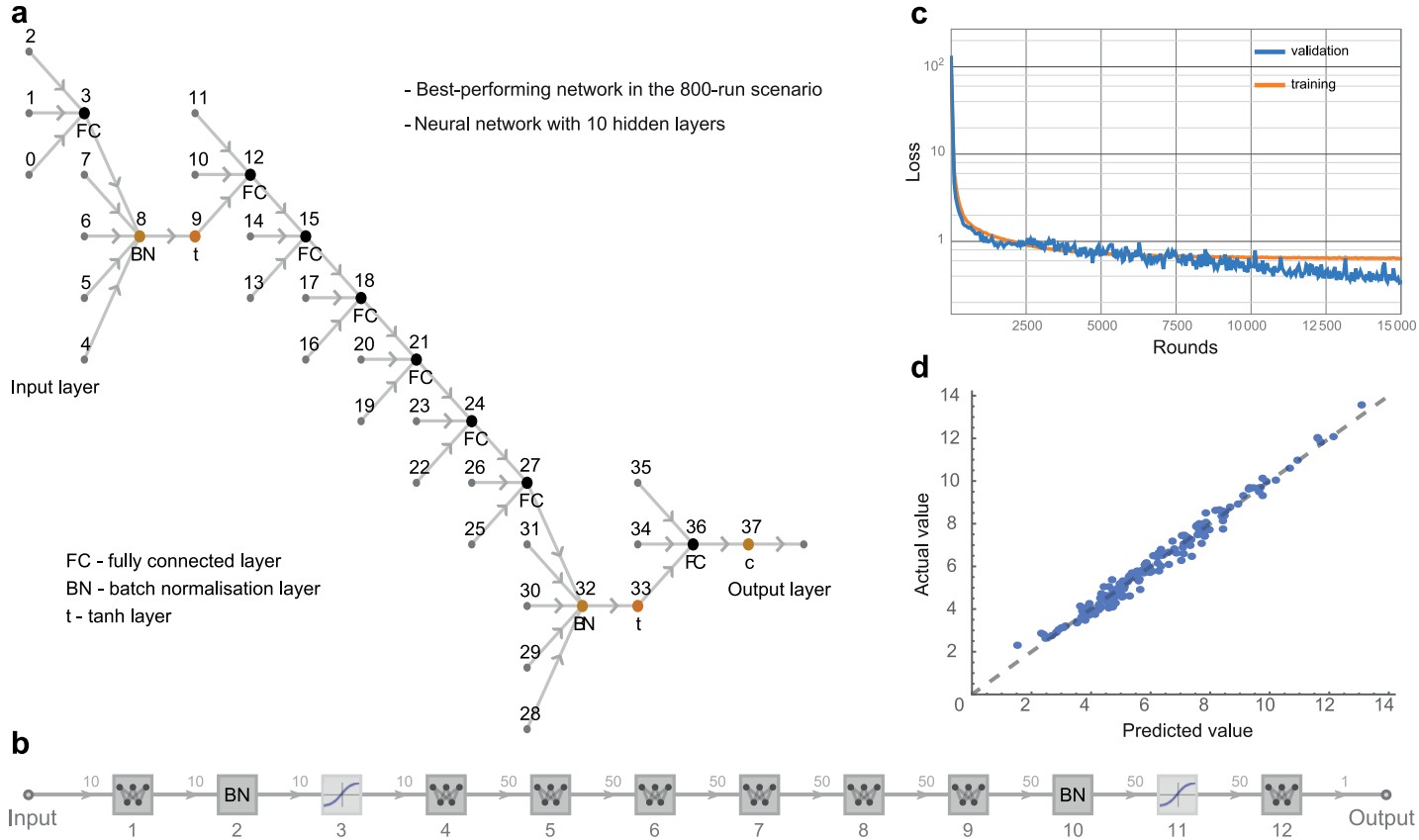

**Fig 2. Sample results on the 800-run simulation scenario.** Diagrams of the neural network architecture in full detail in (a) and in simplified schematic form in (b). In the 800-run scenario, the network with 10 hidden layers pictured here performed the best in a brief comparison between networks with varying numbers of hidden layers. (c) Loss of a 15000-round training run of the simple neural network. (d) Comparison plot produced after training the neural network on the simulation data.

## Investigation of the simulation outputs

In order to further investigate the contributions of the simulation parameters to the variance of the final ABM output, we used Principal Component Analysis (PCA). The variables accounting for the greatest variation in per capita social care cost are reported in Fig 4. PCA is a widely used technique used to determine the variables causing the largest variation in a dataset [51]. To ensure that the directions of maximum variation were not affected by the different magnitudes of the variables, the data was first normalised using the $z$-score:

$$z = \frac{x - \bar{x}}{\sigma}, \tag{2}$$

where $x$ is the raw score, $\bar{x}$ is the sample mean, and $\sigma$ is the standard deviation.

The significant principal components from the analysis were selected according to the Kaiser criterion [52] and Jolliffe's rule [53]. We therefore retained any component that has an eigenvalue with a magnitude greater than one, while additionally requiring that 70% of the variation must be explained, which may require the addition of more components. Applying these criteria to the datasets containing 200 and 400 samples, PCA reduces the dimensionality of the data to seven principal components. PCA identifies a single component in the datasets containing 800 and 1600 samples.

(a)

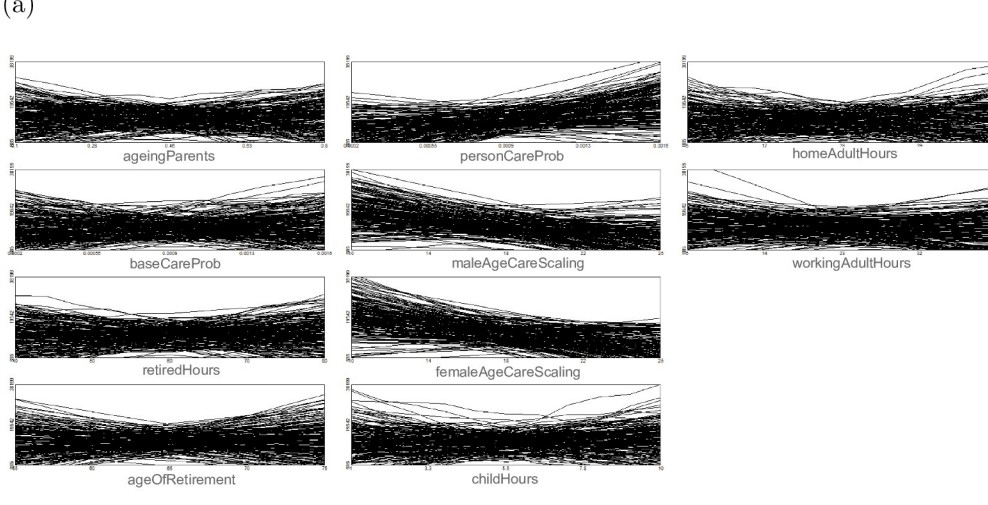

(b)

| Output quantity | Value |
|---|---|
| Largest standardised error | 0.1147 |
| Cross-validation variance range | $4.9762 - 5.5190 \times 10^9$ |
| Estimate of mean output | 11892.8 |
| Estimate of output variance | $1.4828 \times 10^7$ |
| Estimate of total output variance | $5.4145 \times 10^9$ |

**Fig 3. Output of the GP emulator run, performed using the 400-run simulation data set.** (a) Graphs of the main effects of each of the 10 input parameters on the final output of interest, in this case social care cost per person per year. The graphs demonstrate that the emulator was unable to fit a model to the simulation results, as each successive emulator run produced very different results and estimates of the main effects. (b) Numerical outputs of the emulator. The emulator estimates total output variance at 5.41 billion, a clear indication that the emulator is not able to fit the simulation data.

We then considered the contribution of each variable towards each principal component [54]. To determine the variable(s) that contribute most to the variance of each component, a correlation above 0.5 was considered significant. As the number of samples in the dataset increases, PCA gives increasingly ambiguous results. At 200 and 400 samples, the first and last components are separated by less than 2% and 1% contribution to variance respectively showing very little discrimination between components. Additionally, the two highest contributing variables to the first component in the 200 dataset are different from those identified in the 400 dataset. At 200 samples the highest contributors to principal component 1 are homeAdultHours and ageOfRetirement. Conversely, at 400 samples the highest contributors to principal component 1 are maleAgeCareScaling and workingAdultHours. HomeAdultHours governs the amount of hours unemployed adults are able to devote to care, workingAdultHours determines how many hours working adults can allocate to care, and ageOfRetirement determines when agents are able to retire from work, thus freeing up more hours for care.

These variables being the highest contributors suggests that increased availability of informal care hours for adults, either by increasing their allocation generally or through retirement, has a substantial impact on unmet care need. The parameter maleAgeCareScaling quantifies the amount of care need exhibited in the male agent population as they age, so a large change in this value can significantly impact the general level of unmet care need in the agent

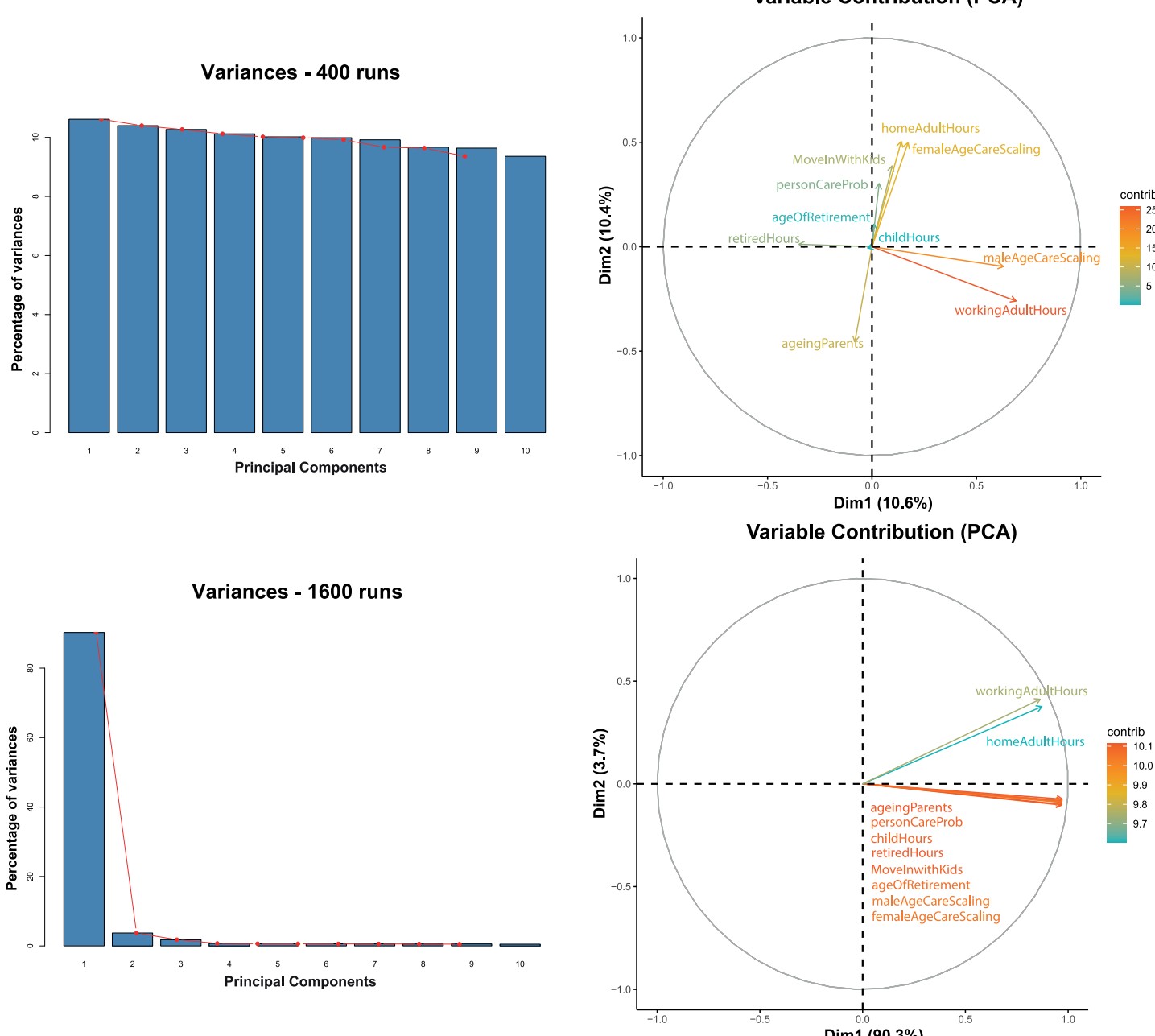

**Fig 4. PCA variable contribution maps and scree plots for the 400- and 1600-sample datasets.** The scree plots of the percent variance contribution of each component visually convey the location where there is a sharp change in gradient, which defines the number of significant components, i.e. the components to be retained in the analysis. The gradient change seen at component 6 of the 400-sample dataset contrasts with the steep gradient change at component 1 of the 1600-sample dataset. The 400-sample dataset variable contribution map shows variables beginning to be clustered, however, there is very little separating the contribution to variance between components with less than 2% difference between the first and last components (as can be seen in the 400-sample scree plot). The variable contribution map of the 1600-sample dataset shows the variables converging into a single component (component 1) contributing 90.3% of the variance. Here PCA is unable to make any useful discrimination between the variables, while identifying eight parameters (on the first component) significantly explaining the variance in the ABM social care per capita.

population. At 800 and 1600 samples, PCA fails to discriminate between the variables and counts all them all as significant contributors to component one, and therefore failing to reduce dimensionality. These results show that PCA is less reliable on smaller datasets, and at 800 samples and above, it is unable to identify which variables have the strongest contribution

to the social care cost per capita. This suggests that once a surrogate has been identified it will require further hyperparameter optimisation and a strategy to reduce overfitting.

## Surrogate modelling results

In total, nine different machine-learning methods were implemented to attempt to replicate the behaviour of the ABM (Fig 1). The PCA results reported above demonstrate the difficulties presented in modelling ABM outputs. More specifically, the parameter spaces in such models tend to be complex, and the contribution of model parameters to output variance is difficult to unravel. We tested each of the nine methods on their ability to replicate the final output of the ABM, comparing the strength of fit and computation time. Each method was tested on four datasets containing the results of 200, 400, 800 and 1600 simulation runs; each dataset examines the same range of the model parameter space at an increasingly higher resolution.

The full results are reported in Fig 1. Neural networks were the strongest performers overall, particularly in the 800- and 1600-sample cases, although this comes at the cost of a computational speed considerably lower than the other methods. Gradient-boosted trees and nonlinear SVM have generally a slightly higher error, but considerably faster runtime. Sample values predicted by each machine learning method are provided in Fig 5 for the 200-run simulation scenario. The results are also shared in full in our GitHub repository https://github.com/thorsilver/Emulating-ABMs-with-ML.

**Linear regression.** The linear regression results further reinforce the PCA results, namely that the ABM behaviour is difficult to approximate with linear models. The MSE loss in the 200 and 400 sample scenarios was significantly higher than the best-performing methods, 5.237 at 200 samples and 4.374 at 400 samples, and at 800 and 1600 samples MSE loss was worst and the second-worst performer (13.439 and 24.18, respectively). This suggests that linear regression was not able to adequately capture the complex behaviour of the original ABM. This is typical of many agent-based modelling studies, where non-linear and even chaotic behaviour is frequently observed.

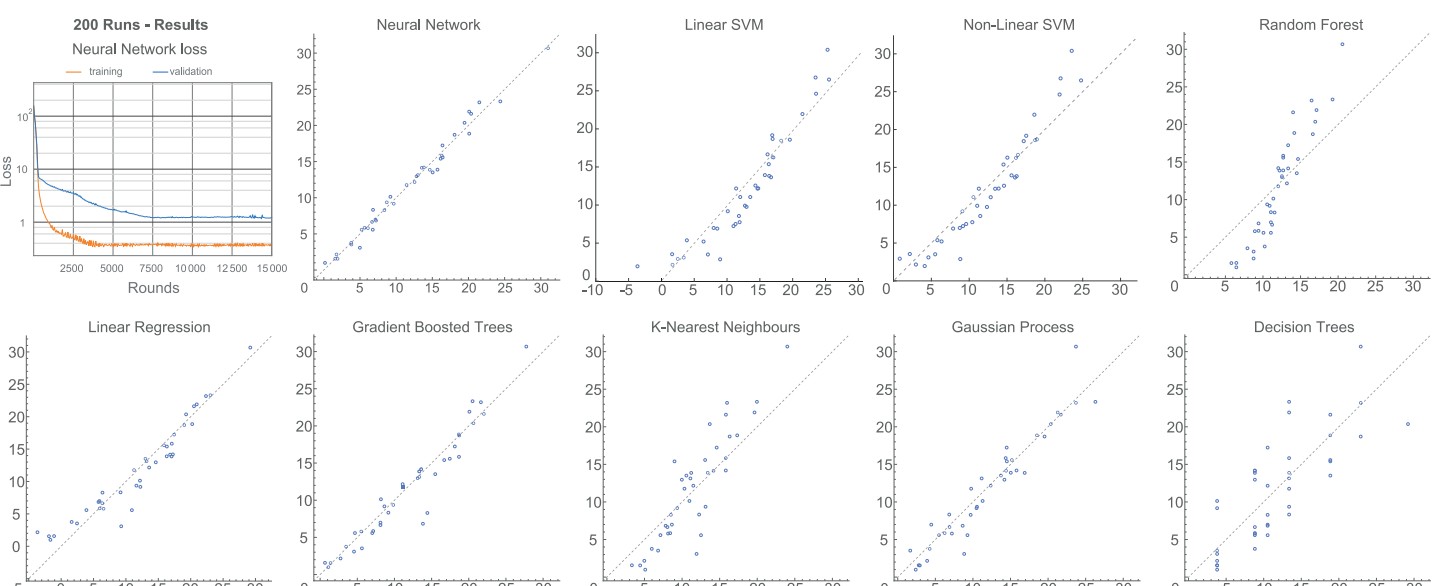

**Fig 5. Predicted value ($x$ axis) versus actual value ($y$ axis) for the 200 run scenario, across all the methods implemented in our comparative study.** The dotted line represents the $y = x$ line.

**Decision trees.** Decision trees, in contrast to linear regression, showed increasing accuracy as the number of observations increased. Decision trees in the 200 and 400 sample scenarios were the second-worst-performing out of the nine methods tested, with an MSE loss of 19.734 and 14.321 at 200 and 400 samples respectively. In the 800 sample scenario, the MSE loss of 3.855 still placed decision trees as the third-worst performer behind linear SVM and linear regression. At 1600 samples, the predictive performance is inadequate as minimising the loss induces the algorithm to predict a constant output for all input parameters. Training times were among the fastest of the methods tested. Decision trees are fast to train and highly interpretable, and are therefore a popular method for classification problems; however, they are less well-suited to problems featuring continuous variables, which may have contributed to these results. As seen in the Random Forest and Gradient-Boosted Trees results below, bagging and boosting can produce significant improvement in performance in some cases.

**Random forest.** Similar to decision trees, Random Forest performed less well in the scenarios with lower numbers of samples, producing the second-worst (16.936) and third-worst (12.06) MSE loss results at 200 and 400 samples. At 800 and 1600 samples, Random Forest produced substantially lower MSE loss but still lower than the strongest-performing methods, with MSE loss of 2.656 at 800 samples and 4.31 at 1600 samples. Similar to Decision Trees, the reported MSE loss is not an ideal loss function in both the 800- and 1600-run scenarios. Training times were low and comparable with gradient-boosted trees.

**Gradient-boosted trees.** Gradient-boosted trees were among the more consistent performers, ranking in the middle of the table at 200 and 400 samples (with MSE loss of 3.778 and 2.526 respectively). The 800 and 1600 sample scenarios showed gradient-boosted trees producing the third and fourth-lowest MSE loss out of the nine methods tested (0.452 and 2.774, respectively). However, as with the other tree-based algorithms, in the 800-run scenario the Gradient-boosted trees show poor predictive performance. Overall, our results suggest that gradient-boosted trees are relatively strong performers on this task, with good performance in three scenarios and quick training times, but the inconsistency demonstrated in the 1600-run scenario shows that the results must be interpreted cautiously. Alternatively, a multitude of loss functions should be used and evaluated when training tree-based methods in this context. Overall, the results show that the Gradient-boosted Trees largely improve upon the performance of Decision Trees and Random Forest.

**K-Nearest neighbours.** K-Nearest Neighbours (KNN) followed a familiar pattern to the other methods, showing high MSE loss at the 200, 400 and 1600 samples (12.895, 13.422, and 7.404 respectively), then a significantly lower MSE loss at 800 samples (1.356). In the case of KNN, we observe a small difference, with the 200/400/1600 sample results ranking slightly better against the other methods than the 800 sample results. We note that many of the nine methods showed improved performance at 800 samples, so even with the marked reduction in MSE loss, for KNN the actual improvement in ranking was small. Training times were among the fastest of the methods tested.

**Gaussian process emulation.** As expected given the particular characteristics of Gaussian Processes (GPs), the method produced relatively strong results when applied to smaller datasets (MSE loss of 4.284 at 200 samples, 1.928 at 400 samples, 0.33 at 800 samples), and very poor results at high numbers of observations (MSE loss of 37.02 at 1600 samples, the worst performer of the 9 methods). Training times were relatively long compared to the other methods tested, with the second-slowest training times of the methods tested; however, the gap between GPs and the slowest method, neural networks, was significant. These results confirm that GPs are powerful and useful for emulation of many varieties of complex computer simulations, but can prove less effective with ABMs, where the outputs do not necessarily vary smoothly with small changes in input parameters.

These results conform with expectations, given the core assumptions underlying the method, namely that the model to be approximated is a smooth, continuous function of its inputs—conditions not met by the vast majority of models [13]. As shown by our initial analysis of this model, the simulation outputs are complex and violate this core assumption. When GP emulation is applied to simulations with sharp changes in outputs due to non-linear processes or sudden phase changes, the emulator will smooth out these spikes, producing very poor results in that area of the parameter space. However, when a model does generate outputs that vary smoothly and continuously with input parameter changes, GP emulation is likely to perform significantly better than in this case study.

**Support vector machine.** Two methods of SVM were used, linear SVM and non-linear (Gaussian) SVM. Non-linear SVM consistently outperformed Linear SVM across all sample sizes, reinforcing that linear models struggle to be valid surrogate models of ABMs. The MSE loss for non-linear SVM decreased as the sample size increased with the lowest loss achieved at 1600 samples (0.236) second in performance only to neural networks (0.188). The performance of linear SVM dipped with the 400 and 800 samples sizes (7.023 and 7.826 respectively) and then began to improve again at the 1600 sample size (3.22), although still significantly worse than the best performers. Non-linear SVM has a relatively fast runtime compared to neural networks, making it a competitive performer.

**Neural networks.** Neural networks proved to be highly consistent performers across all four scenarios, with very low MSE loss recorded across all four scenarios. They were the third-lowest (0.965) at 200 samples, second-lowest at 400, and outperformed all eight rival methods in the 800 sample (0.224) and 1600 sample (0.188) scenarios. However, their training time was orders of magnitude longer than any of the other eight methods, particularly compared to SVM, Decision Trees, KNN and Random Forest. We note that, in this particular comparison, we trained the neural networks on CPU to allow for a direct comparison of runtime with the other methods. In practice, however, using GPUs would speed up training times considerably, making them the strongest candidate method for the emulation of ABMs.

## A note on model assessment

Throughout this study, we have relied on MSE metrics and predicted-versus-actual comparison plots as our primary measures of the performance of each of the chosen methods. Other metrics like the mean absolute error (MAE) or the coefficient of determination [55], or a combination of multiple metrics could be used [56].

Machine-learning methods may achieve results that appear reasonable even when the model does not converge. Especially after normalisation, some methods may appear to perform well under certain metrics. Therefore, the additional context provided by multiple metrics will help ensure that an appropriate evaluation of model performance is made even in those circumstances. In this study, the MSE results provide a measure of performance, which is then supplemented and contextualised by the comparison plots (predicted versus actual value) for each method, in order to provide a straightforward comparison between model predictions and actual values.

## Discussion

We tested a range of machine learning tools towards replicating the outcome of a full ABM. Our results suggest that a deep learning approach (using multi-layered neural networks) is the most promising candidate to create a surrogate of the ABM. Neural networks can be trained on the data deriving from the ABM simulations, and can then replicate the ABM output with high accuracy. In general, we have shown that machine learning methods are able to

approximate ABM predictions with high accuracy and dramatically reduced computational effort, and thus have great potential as a means for more efficient parameter calibration and sensitivity analysis.

The results in Fig 1 also illustrate the challenges of surrogate modelling for ABMs. The performance of most methods—with the notable exception of the neural networks—vary significantly depending on the number of observations. Linear regression, for example, performs well at 200 and 400 runs, but fails to produce a good fit at 800 and 1600, where the complex and jagged nature of the model parameter space becomes more evident. The Gaussian process emulation runs are also instructive—GPs cannot produce a good fit when the function being emulated does not vary smoothly, as is the case in the 1600-run scenario.

Neural networks, by contrast, improve steadily in predictive accuracy as the number of observations increases. As per the universal approximation theorem, neural networks with non-linear activation functions have been shown to be able to approximate any continuous function to an arbitrary degree of precision [57]. In our case, they can accurately approximate our simulation output despite its non-linearity and lack of smoothness. There is a significant computational cost, however, in that a more accurate approximation of more complex functions necessitates exponentially more neurons, which leads to longer training times; these shortcomings are reflected in the large increase in runtime for the neural networks as the numbers of observations increase.

In our testing for this paper, we decided to perform all the machine-learning tasks using consumer-class CPUs only, to allow for comparisons of computation times. Arguably, this leaves neural networks at a disadvantage, as training neural networks on GPUs leads to enormous reductions in runtime. In this case, the model was moderately complex, meaning that even the 11-layer network used for the 1600-run scenario could be trained in a matter of minutes. For surrogate modelling of more complex ABMs, however, training the neural networks on GPUs will significantly reduce their training time.

We should note that the complexity of the neural network architecture could be increased—all networks used here are stacks of fully connected layers with sigmoid activation functions. Despite the lack of extensive architectural optimisation, the neural network models are very accurate in all four scenarios, in some cases significantly more than the other methods. More optimised architectures could potentially achieve even greater accuracy, but at a cost in terms of testing and hyperparameter optimisation. Ultimately, a balanced approach could prove the most effective when approximating ABMs of medium complexity, as our results suggest that even simplistic neural networks can perform very well as surrogate models.

Out of the other machine-learning methods tested, non-Linear (Gaussian) SVM, gradient-boosted trees and Gaussian process emulation were the next strongest performers. However, Gradient-boosted trees proved to be somewhat inconsistent, and showed poor predictive performance in the 1600-run scenario, suggesting that some caution may be warranted when using this method, and that hyperparameter tuning may be necessary to achieve optimal results.

Non-linear SVM was also a competitive method, improving in performance as the sample size increased, and becoming the second-best performer in the 1600-sample. The Gradient-Boosted Trees performed reasonably well in three scenarios, while the GPs were particularly strong in the 800-run scenario and particularly weak in the 1600-run scenario. Using highly-optimised boosted-tree implementations like XGBoost and LightGBM, although unsuccessful in our model, may prove efficient for more complex ABMs, as they are specialised for use on very large datasets.

While this work provides a useful comparison of machine-learning-based surrogate modelling methods applied to a moderately complex ABM, further work could expand on these comparisons. In our experiments, our surrogate models were only required to approximate the behaviour of a model with a single major output of interest; future work could examine how these methods compare when applied to models with multiple outputs. Some of the methods demonstrated here may show improved performance with larger numbers of training examples; however, given that the primary benefit of surrogate modelling is to reduce computation time, we chose to investigate how these methods performed in more constrained scenarios. Future work may supplement this proof-of-concept exploration with more detailed studies of particular machine-learning approaches, with larger numbers of simulation runs produced for training.

Furthermore, alternative models for emulation, e.g. Kriging, could be tested and cross-compared with all methods. For instance, when predicting the surface of the Island growth model, Lamperti et al. [23] showed that Kriging was clearly outperformed by machine learning surrogates (XGBoost in particular) in terms of precision and computational efficiency. In our Linked Lives case study, gradient boosted trees perform consistently well, but they are outperformed by neural networks. For these reasons, we preferred to keep the comparison open as the results might differ depending on the type and complexity of ABM under investigation. Gaussian process emulation has been used for emulation of the simulations with multiple outputs, though generating experimental designs is more complicated in this context [58]. There may also be circumstances where a surrogate model that replicates simulation behaviour over time is desirable, rather than focussing only on the final outputs at the end of a simulation run. In this case, machine-learning methods used for predicting time-series data, such as recurrent neural networks, could serve as useful surrogate models.

Given the diversity of agent-based modelling approaches that have been applied across numerous disciplines, providing a comprehensive comparison of machine-learning-based surrogate modelling methods would be impractical. Conversely, we aimed to provide a representative example of how these methods may be applied in practice, particularly in policy-relevant modelling areas where time and resources are limited. Our results demonstrate that in these challenging environments, machine-learning-based emulation practices may be applied successfully even with minimal hyperparameter optimisation.

While we cannot make a definitive statement on which method of those examined here will prove most reliable in agent-based modelling applications overall, our results suggest that when faced with a model displaying complex behaviour that is not amenable to a Gaussian process emulation approach, neural networks and gradient-boosted trees may be viable replacements. Relatively simple neural network surrogates can be trained effectively even on consumer-level GPUs, but for more complex applications GPU-based training may be necessary. For time-critical applications, or when computational power is at a premium, gradient-boosted trees may be a suitable alternative, given their relatively consistent performance and quick training times even on commodity hardware.

## Conclusion

Gaussian Process emulation has become a popular method for analysing simulation outputs due to their flexibility, power, and the useful measures of uncertainty they can produce. However, their limitations suggest that they are less useful for analysing some complex agent-based simulations. Our comparisons here have shown that alternative ML methods can be viable alternatives in these circumstances.

While these alternative ML methods are promising for this application, work remains to be done to establish standards and practices when using surrogate models in this way. As shown

in our results, each method can display significant differences in performance as more training examples are provided, and given the tendency of ABMs to generate unexpected emergent behaviour, deciding a suitable cut-off point when generating simulation outputs is a challenging problem. Future work will need to examine more case studies in order to develop suitable guidance and best practices when using these methods.

Before these alternative surrogate modelling methods can become widely adopted, we must find efficient and effective ways of making use of these more accurate surrogate models. Unlike GPs, methods like neural networks, non-linear SVM and gradient-boosted trees do not bring with them insightful uncertainty quantification measures 'for free'. In this respect, more effort will need to be spent on the interpretation of the surrogates, which can be challenging when the surrogate models are complex.

However, we note that the often-repeated notion that neural networks or other ML approaches are less interpretable than linear methods is not always clear in practice [29]. Therefore, we avoided classifying these methods by interpretability, as the concept itself is poorly defined and its relative importance will vary enormously depending on the application, whereas runtimes and performance are more broadly useful measures. In addition, as mentioned above, even an opaque surrogate can enable powerful analyses to be performed more quickly, and thus aid in the interpretation of the original simulation.

Interpretable machine learning has become a field in its own right in recent years, and powerful techniques for interpretable ML models are now widely accessible [59, 60]. As specialised libraries for ML-based surrogate modelling become available, interpretability tools can be integrated into the process, enabling modellers to generate additional understanding from their surrogates as well as more rapid sensitivity analyses. In later extensions of this work, we aim to investigate the use of these tools in ML-based surrogate modelling of agent-based simulations.

Following on from this study, we also aim to focus on developing libraries and examples of how to use ML-based surrogate models to generate deeper insights into the behaviour of complex ABMs. This may significantly enhance future modelling efforts in the ABM community by increasing the accessibility of these techniques. These additional methods for simulation analysis would serve to fill the gaps where GPs are unable to produce usable surrogates, ensuring that detailed sensitivity and uncertainty analysis become the norm in agent-based modelling studies.

## Author Contributions

**Conceptualization:** Claudio Angione, Eric Silverman.

**Data curation:** Eric Silverman, Elisabeth Yaneske.

**Formal analysis:** Claudio Angione, Eric Silverman.

**Funding acquisition:** Claudio Angione, Eric Silverman.

**Investigation:** Claudio Angione, Eric Silverman, Elisabeth Yaneske.

**Methodology:** Claudio Angione, Eric Silverman.

**Project administration:** Claudio Angione, Eric Silverman.

**Software:** Eric Silverman, Elisabeth Yaneske.

**Supervision:** Claudio Angione.

**Validation:** Eric Silverman.

**Visualization:** Claudio Angione, Eric Silverman, Elisabeth Yaneske.

**Writing – original draft:** Claudio Angione, Eric Silverman, Elisabeth Yaneske.

**Writing – review & editing:** Claudio Angione, Eric Silverman, Elisabeth Yaneske.

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
