## [Decision Letter · Decision Letter 0]

22 Sep 2021

PONE-D-21-24064Using Machine Learning to Emulate Agent-Based SimulationsPLOS ONE

Dear Dr. Angione,

Thank you for submitting your manuscript to PLOS ONE. After careful consideration, we feel that it has merit but does not fully meet PLOS ONE’s publication criteria as it currently stands. Therefore, we invite you to submit a revised version of the manuscript that addresses the points raised during the review process.

We look forward to receiving your revised manuscript.

Kind regards,

Roland Bouffanais, Ph.D.

Academic Editor

PLOS ONE

Journal Requirements:

"ES is part of the Complexity in Health Improvement Programme supported by the Medical Research Council (MC UU 00022/1) and the Chief Scientist Office (SPHSU16). CA would like to acknowledge the support from UKRI Research England’s THYME project, and from the Children’s Liver Disease Foundation. This work was supported by UK Prevention Research Partnership MR/S037594/1, which is funded by the British Heart Foundation, Cancer Research UK, Chief Scientist Office of the Scottish 

Government Health and Social Care Directorates, Engineering and Physical Sciences Research Council, Economic and Social Research Council, Health and Social Care Research and Development Division (Welsh Government), Medical Research Council, National Institute for Health Research, Natural Environment Research Council, Public Health Agency (Northern Ireland), The Health Foundation and Wellcome"

"project, and from the Children's Liver Disease Foundation.  This work was supported by UK Prevention Research Partnership MR/S037594/1, which is funded by the British Heart Foundation, Cancer Research UK, Chief Scientist Office of the Scottish Government Health and Social Care Directorates, Engineering and Physical Sciences Research Council, Economic and Social Research Council, Health and Social Care Research and Development Division (Welsh Government), Medical Research Council, National Institute for Health Research, Natural Environment Research Council, Public Health Agency (Northern Ireland), The Health Foundation and Wellcome.

The funders had no role in study design, data collection and analysis, decision to publish, or preparation of the manuscript"

Additional Editor Comments (if provided):

Both reviewers highlight the fact that your manuscript has merit and that it deals with an important topic.

However, as you'll read below, both reviewers expressed some concerns and felt that some points should be addressed before this manuscript can be published.

We look forward to receiving your revised manuscript.

Reviewers' comments:

Reviewer's Responses to Questions

**Comments to the Author**

1. Is the manuscript technically sound, and do the data support the conclusions?

Reviewer #1: Partly

Reviewer #2: Yes

2. Has the statistical analysis been performed appropriately and rigorously? 

Reviewer #1: Yes

Reviewer #2: N/A

3. Have the authors made all data underlying the findings in their manuscript fully available?

Reviewer #1: Yes

Reviewer #2: Yes

4. Is the manuscript presented in an intelligible fashion and written in standard English?

Reviewer #1: Yes

Reviewer #2: Yes

5. Review Comments to the Author

Reviewer #1: The authors propose a careful and detailed comparison of machine-learning surrogates emulating the behavior of an agent-based model designed to evaluate social care policies in the UK. The authors consider both accuracy of prediction and efficiency (i.e. speed). The use of emulators is sometimes the only way to perform sensitivity and calibration of ABMs: therefore, comparing various approaches and disclosing the codes needed to obtain such emulators is relevant. In this respect, the paper is very welcome.

Indeed, I feel that, upon adequate revision, the paper has the potential to make a solid contribution to the social simulation/ABM literature.

Below I provide my comments, not in order of importance

1. The authors claim that machine-learning methods have rarely been employed in the ABM literature (pag. 3). This is partially true, but I suggest them looking at the recent survey by Dahlke et al 2020. In addition, please note that Lamperti et al. 2018, which the authors cite, conduct an exercise which is similar to the one proposed in this paper by comparing surrogate models obtained with GP and Boosted Trees (through XGBoost) across different metrics and using various sample sizes. I suggest the authors to discuss their results in light of those obtained in Lamperti et al. 2018 and in the rest of the literature. I feel this paper consolidates - and substantially expand - some evidence which was already there: this is a value added.

2. One crucial element in the analysis of stochastic ABMs embedding a time component is ergodicity, which I intend here as the presence of a single statistical equilibrium to which various runs (i.e. model runs with different seeds and same parameter configuration) approximately converge after a warm-up period (see for example Grazzini 2012; Vandin et al. 2021). My feeling is that surrogate modeling needs ergodicity as a prerequisite. It seems to me that the model considered by the author is deterministic (apart from the choice of initial conditions); however, several ABMs embed random draws during the simulation, and I feel the authors need to discuss whether their exercise and results can be extended to this class of models and under which conditions.

3. Is the interpretability of the ML surrogate valuable? The authors acknowledge that surrogates might be useful not just to improve the efficiency of simulations (which is necessary for - e.g. - global sensitivity analysis), but to help understand the model behavior, which is often complex (pag.4). However, when discussing their results, they do not mention the issue of interpretability, which I feel as relevant instead. For example, decision trees or random forests might help detect which parameters are more relevant to obtain behaviors aligned with the actual model or to models’ fit with the empirical data, therefore helping understand the model itself and its “sensitive” parts. To the eyes of a modeler this is fundamental for calibration and sensitivity analysis. To the contrary, ANNs can difficulty offer such information. I would suggest the authors to build on their observation of pag. 4 and discuss this issue in their Discussion and Conclusions sections.

4. Figure 3a is relevant. It shows that GP emulators (as implemented by the authors) have difficulties. I feel it would be relevant to show (i) the ground truth to which the emulator should be compared and (ii) similar charts for the other surrogates. Also, I would add a note of caution; if the ground truth behavior (i.e. how the actual ABM responds to parameter) is relatively smooth or even linear, learning should be easier. I wonder how the different methods perform when facing highly non-linear model dependence on parameters. I suggest providing – at least - some discussions.

5. This is probably the main concern I have. I have issues at understanding why the accuracy of surrogate models - e.g. GP and Boosted trees - vary so largely when increasing sample size, especially from 800 samples to 1600 samples. The performance seems rather consistent from 200 to 800 samples, and then it drops quite dramatically for 1600 samples. This is rather counter-intuitive to me. What happens, e.g., at 1200 samples? Can they author provide a convincing explanation of this issue, which greatly affects the results proposed by the paper.

Refs

Dahlke, J., Bogner, K., Mueller, M., Berger, T., Pyka, A., & Ebersberger, B. (2020). Is the juice worth the squeeze? machine learning (ml) in and for agent-based modelling (abm). arXiv preprint arXiv:2003.11985.

Lamperti, F., Roventini, A., & Sani, A. (2018). Agent-based model calibration using machine learning surrogates. Journal of Economic Dynamics and Control, 90, 366-389.

Grazzini, J. (2012). Analysis of the emergent properties: Stationarity and ergodicity. Journal of Artificial Societies and Social Simulation, 15(2), 7.

Vandin, A., Giachini, D., Lamperti, F., & Chiaromonte, F. (2021). Automated and Distributed Statistical Analysis of Economic Agent-Based Models. arXiv preprint arXiv:2102.05405.

Reviewer #2: This paper presents a set of experiments to capture agent-based simulation input-output relationships using machine learning approaches. It does so by simulating a model called Linked Lives with different parameter settings. Many machine learning techniques are compared with the Gaussian process (GP) emulation technique.

This paper is written in a clear language and as a reviewer, I had an easy time following its structure. Sharing code and data is a good practice. Well done! Overall, I would like to see this paper published but there are some additional tasks that need to be conducted in order to get to that level. Here are some major concerns:

1. Most agent-based models are stochastic, meaning each run will lead to different output values even if input values are the same. The approach in this paper does not capture uncertainty generated by multiple runs of the same configuration.

2. would recommend finding the best performing ML technique first. Then, I would compare it with GP and other surrogate modeling techniques. Speaking of other techniques, Kriging is also a popular surrogate modeling technique to compare against.

3. Experiment sample size is quite limited. There is only one ABM used and there is only one output value observed. More ABMs (check Comses) and multi outputs can generate interesting and more comprehensive results.

Here are some minor points:

1. I believe the term “emulate” is not the right one to use in this context. Emulate indicates being “in place of” something and replacing it. In the context mentioned here there is “approximation” not replacement of the real simulation model. Even though the Gaussian process paper uses the term emulator, it is not the right use here. Surrogate model sounds like a better option.

2. The authors define ABM as both “Agent-Based Modelling” and “Agent-Based Model.” These two are not interchangeable. Please select one and use it consistently.

3. The first paragraph in the second page of the article starts describing agent-based modeling without any references. Some introductory references are needed.

4. “When ABMs are highly complex, performing these kinds of analyses becomes both time- and cost-prohibitive, potentially leading some modellers to truncate these analyses or eliminate them entirely.” I would add the implications of this statement which is related to verification and validation.

“ABM-based modelling” redundant words.

5. The paper calls the case study as moderate-complexity ABM. It’s not clear how model complexity is measured. Are there any objective techniques to do so?

6. Why spider plot :) Barplot would work better.

7. In figure 5, is the dotted line y=x? If so, please mention.

6. PLOS authors have the option to publish the peer review history of their article (what does this mean?). If published, this will include your full peer review and any attached files.

Reviewer #1: No

Reviewer #2: **Yes: **Hamdi Kavak

---

## [Author Response · Author response to Decision Letter 0]

13 Dec 2021

Please find attached our point-by-point response.

---

## [Decision Letter · Decision Letter 1]

13 Jan 2022

Using Machine Learning as a Surrogate Model for Agent-Based Simulations

PONE-D-21-24064R1

Dear Dr. Angione,

We’re pleased to inform you that your manuscript has been judged scientifically suitable for publication and will be formally accepted for publication once it meets all outstanding technical requirements.

Kind regards,

Roland Bouffanais, Ph.D.

Academic Editor

PLOS ONE

Additional Editor Comments (optional):

Reviewers' comments:

Reviewer's Responses to Questions

**Comments to the Author**

1. If the authors have adequately addressed your comments raised in a previous round of review and you feel that this manuscript is now acceptable for publication, you may indicate that here to bypass the “Comments to the Author” section, enter your conflict of interest statement in the “Confidential to Editor” section, and submit your "Accept" recommendation.

Reviewer #2: All comments have been addressed

2. Is the manuscript technically sound, and do the data support the conclusions?

Reviewer #2: Yes

3. Has the statistical analysis been performed appropriately and rigorously? 

Reviewer #2: Yes

4. Have the authors made all data underlying the findings in their manuscript fully available?

Reviewer #2: Yes

5. Is the manuscript presented in an intelligible fashion and written in standard English?

Reviewer #2: Yes

6. Review Comments to the Author

Reviewer #2: Authors have done a great job in addressing the reviewer comments. I have no further concerns about the paper. Well done.

7. PLOS authors have the option to publish the peer review history of their article (what does this mean?). If published, this will include your full peer review and any attached files.

Reviewer #2: No

---

## [Editor Report · Acceptance letter]

21 Jan 2022

PONE-D-21-24064R1 

Using Machine Learning as a Surrogate Model for Agent-Based Simulations 

Dear Dr. Angione:

I'm pleased to inform you that your manuscript has been deemed suitable for publication in PLOS ONE. Congratulations! Your manuscript is now with our production department. 

Kind regards, 

on behalf of

Professor Roland Bouffanais 

Academic Editor

PLOS ONE